# Probabilistic Pairwise Model Comparisons Based on Bootstrap Estimators of the Kullback–Leibler Discrepancy

**DOI:** 10.3390/e24101483

**Published:** 2022-10-18

**Authors:** Andres Dajles, Joseph Cavanaugh

**Affiliations:** Department of Biostatistics, University of Iowa, 145 N. Riverside Drive, Iowa City, IA 52242, USA

**Keywords:** bootstrap discrepancy comparison probability (BDCP), discrepancy comparison probability (DCP), likelihood ratio test (LRT), model selection, *p*-value

## Abstract

When choosing between two candidate models, classical hypothesis testing presents two main limitations: first, the models being tested have to be nested, and second, one of the candidate models must subsume the structure of the true data-generating model. Discrepancy measures have been used as an alternative method to select models without the need to rely upon the aforementioned assumptions. In this paper, we utilize a bootstrap approximation of the Kullback–Leibler discrepancy (BD) to estimate the probability that the fitted null model is closer to the underlying generating model than the fitted alternative model. We propose correcting for the bias of the BD estimator either by adding a bootstrap-based correction or by adding the number of parameters in the candidate model. We exemplify the effect of these corrections on the estimator of the discrepancy probability and explore their behavior in different model comparison settings.

## 1. Introduction

Hypothesis testing and *p*-values are routinely used in applied, empirically oriented research. However, practitioners of statistics often misinterpret *p*-values, particularly in settings where hypothesis tests are used for model comparisons. Riedle, Neath and Cavanaugh [1] attempt to address this issue by providing an alternate conceptualization of the *p*-value. The authors introduce and investigate the concept of the discrepancy comparison probability (DCP) and its bootstrapped estimator, called the bootstrap discrepancy comparison probability (BDCP). The authors establish a clear connection between the BDCP based on the Kullback–Leibler discrepancy (KLD) and the *p*-values derived from likelihood ratio tests. However, this connection only exists when using the bootstrap discrepancy (BD) that arises from the “plug-in” principle, which yields a biased approximation to the KLD. Similarly to complexity penalization of the Akaike Information Criterion (AIC), we establish that an intuitive bias correction to the BD is the addition of *k*, the number of functionally independent parameters in the candidate model. We also propose utilizing a bootstrap-based correction, which can be justified under less stringent assumptions. We analyze how well the bootstrap approach corrects the bias of the BDCP and the BD, and we show that, in most settings, its performance is comparable to simply adding *k*.

## 2. Methodological Development

### 2.1. Background

When faced with the task of choosing amongst competing models, statisticians often use discrepancy or divergence functions. One of the most flexible and ubiquitous divergence measures is the Kullback–Leibler information. To introduce this measure in the present context, consider a vector of independent observations $y={\left(y_{1},y_{2},\ldots ,y_{n}\right)}^{T}$ such that *y* is generated from an unknown distribution g(y). Suppose that a candidate model f(y|θ) is proposed as an approximation for g(y), and that this model belongs to the parametric class of densities
F=[f(y|θ):θ∈Θ],
where Θ is the parameter space for θ. The Kullback–Leibler information, given by
$$I_{KL}\left(g,\theta \right)=E_{g}\left[\log\frac{g(y)}{f(y|\theta )}\right],$$
captures the separation between the proposed model f(y|θ) and the true data-generating model g(y).

Although not a formal metric, $I_{KL}\left(g,\theta \right)$ is characterized by two desirable properties. First, by Jensen’s inequality, $I_{KL}\left(g,\theta \right)\geq 0$ with equality if and only if g(y)=f(y|θ). Second, as the dissimilarity between g(y) and f(y|θ) increases, $I_{KL}\left(g,\theta \right)$ increases accordingly.

Note that we can write
$$\begin{array}{cc} 2I_{KL}\left(g,\theta \right) & =E_{g}\left[-2\log\left(f\left(y|\theta \right)\right)\right]-E_{g}\left[-2\log\left(g\left(y\right)\right)\right]\\ \; & =E_{g}\left[-2\ell \left(\theta |y\right)\right)]-E_{g}\left[-2\log\left(g\left(y\right)\right)\right], \end{array}$$
where log(f(y|θ))=ℓ(θ|y). In the preceding relation, for any proposed candidate model, the quantity $E_{g}\left[-2\log\left(g\left(y\right)\right)\right]$ is constant. Only the quantity $E_{g}\left[-2\ell \left(\theta |y\right)\right]$ changes across different models, which means it is the only quantity needed to distinguish among various models. The expression
$$d\left(g,\theta \right)=E_{g}\left[-2\ell \left(\theta |y\right)\right)]$$
is known as the Kullback–Leibler discrepancy (KLD) and is often used as a substitute for $I_{KL}\left(g,\theta \right)$.

In practice, the goal is to determine the propriety of fitted models of the form $f(y|\hat{\theta })$, where $\hat{\theta }={\mathrm{argmax}}_{\theta \in \Theta }\;\ell (\theta |y)$. The KL discrepancy for the fitted model is given by
$$d\left(g,\hat{\theta }\right)=E_{g}\left[-2\ell \left(\theta |y\right)\right]{|}_{\theta =\hat{\theta }}.$$

### 2.2. The Discrepancy Comparison Probability and Bootstrap Discrepancy Comparison Probability

Suppose that we have two nested models that are formulated to characterize the sample *y*, and we designate one of the models the null, represented by $\theta_{1}$, and the other model the alternative, represented by $\theta_{2}$. The discrepancies under the fitted null and alternative models are given by $d(g,{\hat{\theta }}_{1})$ and $d(g,{\hat{\theta }}_{2})$, respectively. We can use these discrepancies to define the Kullback–Leibler discrepancy comparison probability (KLDCP), which is given by
$$P=\mathrm{Pr}[d\left(g,{\hat{\theta }}_{1}\right)<d\left(g,{\hat{\theta }}_{2}\right)].$$

The KLDCP evaluates the probability that the fitted null model is closer to the true data-generating model than the fitted alternative. The values of $d(g,{\hat{\theta }}_{1})$ and $d(g,{\hat{\theta }}_{2})$ are calculated from the same sample. For example, a KLDCP of 0.8 means that the fitted null has a smaller discrepancy than the fitted alternative in 80% of the samples drawn from the same distribution and of the same size. The development and interpretation of the KLDCP is presented in depth by Riedle, Neath and Cavanaugh [1].

We can estimate the KLDCP using the bootstrap approximation of the joint distribution of $d(g,{\hat{\theta }}_{1})$ and $d(g,{\hat{\theta }}_{2})$. The bootstrap joint distribution is based on the discrepancy estimators that arise from the “plug-in” principle, as described by Efron and Tibshirani [2], which replaces all the elements of the KLD by their bootstrap analogues. Specifically, we replace *g* by the empirical distribution $\hat{g}$; *y* by the bootstrap sample from $\hat{g}$, which we call $y^{*}$; and finally, $\hat{\theta }$ by the maximum likelihood estimate (MLE) derived under the bootstrap sample $y^{*}$, which we call ${\hat{\theta }}^{*}$. With these replacements, the bootstrap version of the KLD is given by
$$\begin{array}{cc} d(\hat{g},{\hat{\theta }}^{*}) & =E_{\hat{g}}{\left[-2\ell \left(\theta |y\right)\right]|}_{\theta ={\hat{\theta }}^{*}}\\ \; & =\sum_{i=1}^{n}-2\ell_{i}\left({\hat{\theta }}^{*}|y_{i}\right)\;(\mathrm{because}\;\mathrm{each}\;y_{i}\;\mathrm{is}\;\mathrm{independent}.)\\ \; & =-2\ell ({\hat{\theta }}^{*}|y), \end{array}$$
where $\ell_{i}$ represents the contribution to the likelihood based on the *i*th response $y_{i}$.

Now, in order to build a bootstrap distribution, we must draw various bootstrap samples from *y*. Suppose that we draw j=1,2,…,J bootstrap samples, and for each of these samples, we calculate the MLE of θ, which we denote as ${\hat{\theta }}^{*}\left(j\right)$. This allows us to obtain a set of *J* different bootstrap discrepancies; this set is defined as
$$\left\{d(\hat{g},{\hat{\theta }}^{*}\left(j\right)):j=1,\ldots ,J\right\},$$
and these variates can be used to construct the bootstrap analogue of the discrepancy distribution.

Finally, we can extend this procedure to the setting of the null and alternative models. For each bootstrap sample, we calculate ${\hat{\theta }}_{2}^{*}\left(j\right)$ and ${\hat{\theta }}_{1}^{*}\left(j\right)$, which are the bootstrap sample MLEs of $\theta_{2}$ and $\theta_{1}$, respectively. We then compute the discrepancies $d(\hat{g},{\hat{\theta }}_{2}^{*}\left(j\right))$ and $d(\hat{g},{\hat{\theta }}_{1}^{*}\left(j\right))$ for the null and alternative models, respectively. This collection of *J* pairs of null and alternative bootstrap discrepancies defines the set
$$\left\{(d\left(\hat{g},{\hat{\theta }}_{1}^{*}\left(j\right)\right),d\left(\hat{g},{\hat{\theta }}_{2}^{*}\left(j\right)\right)):j=1,\ldots ,J\right\},$$
which characterizes the bootstrap analogue of the joint distribution of $d(\hat{g},{\hat{\theta }}_{1})$ and $d(\hat{g},{\hat{\theta }}_{2}).$ The bootstrap distribution can be utilized to estimate the bootstrap analogue of the DCP, given by
$$P^{*}={\mathrm{Pr}}^{*}\left[d\left(\hat{g},{\hat{\theta }}_{1}^{*}\right)<d\left(\hat{g},{\hat{\theta }}_{2}^{*}\right)\right].$$

By the law of large numbers, we can approximate $P^{*}$ by calculating the proportion of times when $d\left(\hat{g},{\hat{\theta }}_{1}^{*}\left(j\right)\right)<d\left(\hat{g},{\hat{\theta }}_{2}^{*}\left(j\right)\right)$ in the *J* bootstrap samples that were drawn. Thus, if *I* is an indicator function, we can define an estimator of the DCP, which we call the bootstrap discrepancy comparison probability (BDCP), as follows: (1)$$\mathrm{BDCP}=\frac{1}{J}\sum_{j=1}^{J}I\left[d\left(\hat{g},{\hat{\theta }}_{1}^{*}\left(j\right)\right)<d\left(\hat{g},{\hat{\theta }}_{2}^{*}\left(j\right)\right)\right].$$

## 3. Bias Corrections for the BDCP

An important issue that arises in the bootstrap estimation of the KLD is the negative bias of the discrepancy estimators that materializes from the “plug-in” principle. The following lemma establishes and quantifies this bias for large-sample settings under an appropriately specified candidate model.

**Lemma** **1.**
*For a large sample size, assuming that the candidate model subsumes the true model, we have*

$$E_{g}\left\{E_{*}\left[-2\ell \left({\hat{\theta }}^{*}|y\right)\right]\right\}\approx E_{g}\left[d\left(g,\hat{\theta }\right)\right]-k,$$

*where $E_{*}$ is the expectation with respect to the bootstrap distribution, and k is the dimension of the model.*


**Proof.** For a maximum likelihood estimator $\hat{\theta }$, it is well known that for a large sample size and under certain regularity conditions, we have
(2)$${\left(\hat{\theta }-\theta \right)}^{T}I\left(\theta |y\right)\left(\hat{\theta }-\theta \right)∼\chi_{k}^{2},$$
provided that the model is adequately specified. In the preceding, $\chi_{k}^{2}$ denotes a centrally distributed chi-square random variable with *k* degrees-of-freedom.Now, consider the second-order Taylor series expansion of $-2\ell ({\hat{\theta }}^{*}|y)$ about $\hat{\theta }$, which results in
(3)$$-2\ell \left({\hat{\theta }}^{*}|y\right)\approx -2\ell \left(\hat{\theta }|y\right)+{\left({\hat{\theta }}^{*}-\hat{\theta }\right)}^{T}I\left(\hat{\theta }|y\right)\left({\hat{\theta }}^{*}-\hat{\theta }\right).$$By taking the expected value of both sides of (Equation 3) with respect to the bootstrap distribution of ${\hat{\theta }}^{*}$, we obtain
$$\begin{array}{cc} E_{*}\left(-2\ell ({\hat{\theta }}^{*}|y)\right) & \approx -E_{*}\left(2\ell (\hat{\theta }|y)\right)+E_{*}\left({\left({\hat{\theta }}^{*}-\hat{\theta }\right)}^{T}I\left(\hat{\theta }|y\right)\left({\hat{\theta }}^{*}-\hat{\theta }\right)\right)\\ \; & \approx -2\ell (\hat{\theta }|y)+k\;(\mathrm{by}\;\mathrm{the}\;\mathrm{approximation}\;\mathrm{in}\;(2)),\\ \; & =\mathrm{AIC}-k, \end{array}$$
where AIC denotes the Akaike information criterion.Finally, it has been established that if the true model is contained in the candidate class at hand, and if the large sample properties of MLEs hold, then AIC serves as an asymptotically unbiased estimator of the KLD. Thus,
$$\begin{array}{cc} E_{g}\left(E_{*}\left(-2\ell ({\hat{\theta }}^{*}|y)\right)\right) & \approx E_{g}\left(\mathrm{AIC}\right)-k\\ \; & \approx E_{g}\left(d\left(g,\hat{\theta }\right)\right)-k. \end{array}$$□

The preceding expression can be re-written as
$$E_{g}\left(d\left(g,\hat{\theta }\right)\right)\approx E_{g}\left(E_{*}\left(-2\ell ({\hat{\theta }}^{*}|y)\right)\right)+k,$$
which implies that the bias correction *k* must be added to the bootstrap discrepancy in the estimation of the KLD. The BD estimator corrected by the addition of *k* will be called BDk.

Now, focus again on Equation (Equation 3). By subtracting $\left(-2\ell \left(\hat{\theta }|y\right)\right)$ from both sides of the equation, we obtain
(4)$$-2\ell \left({\hat{\theta }}^{*}|y\right)-\left(-2\ell \left(\hat{\theta }|y\right)\right)\approx {\left({\hat{\theta }}^{*}-\hat{\theta }\right)}^{T}I\left(\hat{\theta }|y\right)\left({\hat{\theta }}^{*}-\hat{\theta }\right).$$

As mentioned previously, if the candidate model is adequately specified, then the distributional approximation in (Equation 2) holds true. However, if this model specification assumption is not met, then we can utilize the approximation in (Equation 4) to find a suitable bias correction via the bootstrap. The bootstrap has been used for bias corrections in similar problem contexts [3,4].

By applying the expected value with respect to the bootstrap distribution of ${\hat{\theta }}^{*}$ to both sides of (Equation 4), we obtain
(5)$$E_{*}\left(-2\ell ({\hat{\theta }}^{*}|y)\right)-\left(-2\ell \left(\hat{\theta }|y\right)\right)\approx E_{*}\left({\left({\hat{\theta }}^{*}-\hat{\theta }\right)}^{T}I\left(\hat{\theta }|y\right)\left({\hat{\theta }}^{*}-\hat{\theta }\right)\right).$$

The goal is then to find an approximation of $E_{*}\left(-2\ell ({\hat{\theta }}^{*}|y)\right)-\left(-2\ell \left(\hat{\theta }|y\right)\right)$. Note that by the law of large numbers, we have that when J→∞,
$$\frac{1}{J}\sum_{j=1}^{J}-2\ell \left({\hat{\theta }}^{*}\left(j\right)|y\right)\overset{}{\to }E_{*}\left(-2\ell \left({\hat{\theta }}^{*}|y\right)\right).$$ Thus, for J→∞, we can assert
$$\frac{1}{J}\sum_{j=1}^{J}-2\ell \left({\hat{\theta }}^{*}\left(j\right)|y\right)-\left(-2\ell \left(\hat{\theta }|y\right)\right)\overset{}{\to }E_{*}\left(-2\ell \left({\hat{\theta }}^{*}|y\right)\right)-\left(-2\ell \left(\hat{\theta }|y\right)\right).$$

The preceding result shows that $\frac{1}{J}\sum_{j=1}^{J}-2\ell \left({\hat{\theta }}^{*}\left(j\right)|y\right)-\left(-2\ell \left(\hat{\theta }|y\right)\right)$ serves as an asymptotically unbiased estimator of $E_{*}\left(-2\ell \left({\hat{\theta }}^{*}|y\right)\right)-\left(-2\ell \left(\hat{\theta }|y\right)\right)$. We therefore propose using
$$k_{b}=\frac{1}{J}\sum_{j=1}^{J}-2\ell \left({\hat{\theta }}^{*}\left(j\right)|y\right)-\left(-2\ell \left(\hat{\theta }|y\right)\right)$$
as a bootstrap-based correction of the BD. A more in-depth derivation and exploration of the $k_{b}$ correction can be found in Cavanaugh and Shumway [5].

Subsequently, the bootstrap approximation of the KLD with a bootstrap-based bias correction is expressed by $E_{*}\left(-2\ell \left({\hat{\theta }}^{*}|y\right)\right)+k_{b}$, and is estimated by
$$\mathrm{BDb}=\frac{1}{J}\sum_{j=1}^{J}-2\ell \left({\hat{\theta }}^{*}\left(j\right)|y\right)+k_{b}.$$ It follows that the bootstrap bias-corrected BDCP would be defined as
(6)$$\begin{array}{c} \mathrm{BDCPb}=\frac{1}{J}\sum_{j=1}^{J}I\left[d\left(\hat{g},{\hat{\theta }}_{1}^{*}\left(j\right)\right)+k_{1b}<d\left(\hat{g},{\hat{\theta }}_{2}^{*}\left(j\right)\right)+k_{2b}\right], \end{array}$$
where $k_{1b}$ and $k_{2b}$ correspond to the bootstrap-based corrections for the null and alternative models, respectively.

Similarly, the *k* bias-corrected BD is expressed as
$$\mathrm{BDk}=\frac{1}{J}\sum_{j=1}^{J}-2\ell \left({\hat{\theta }}^{*}\left(j\right)|y\right)+k,$$
and the *k* bias-corrected BDCP is given by
(7)$$\begin{array}{c} \mathrm{BDCPk}=\frac{1}{J}\sum_{j=1}^{J}I\left[d\left(\hat{g},{\hat{\theta }}_{1}^{*}\left(j\right)\right)+k_{1}<d\left(\hat{g},{\hat{\theta }}_{2}^{*}\left(j\right)\right)+k_{2}\right], \end{array}$$
where $k_{1}$ and $k_{2}$ are the number of functionally independent parameters that define the null and alternative models, respectively.

## 4. Simulation Studies

The following simulation sets are designed to explore the bias when estimating both the DCP based on the Kullback–Leibler discrepancy (KLDCP) and the expected value of the KLD. We present different hypothesis testing scenarios, not all of which are conventional, under a linear data-generating model and for varying sample sizes. Each setting exhibits three different approaches to formulating the BD: adding the bootstrap-based correction (BDb), adding *k* (BDk), and leaving the estimator uncorrected.

### 4.1. Settings for Simulation Sets

For Sets 1 to 5, the true data-generating model is of the form
$$y_{i}=x_{i}^{T}\beta_{0}+\epsilon_{i},$$
with $\beta_{0}^{T}=\left[\begin{array}{cccc} \beta_{0,1} & \beta_{0,2} & \ldots & \beta_{0,p} \end{array}\right],$
$x_{i}^{T}=\left[\begin{array}{cccc} 1 & x_{i2} & \ldots & x_{ip} \end{array}\right],$ and
(8)$${\left[\begin{array}{ccc} x_{i2} & \ldots & x_{ip} \end{array}\right]}^{T}∼N_{p-1}\left(\mu ,\Sigma \right),$$
where the entries of μ are chosen from {−1,1} with equal probability, and $\Sigma =diag_{p-1}\left(100\right)$. For Sets 1 to 4, we have $\epsilon_{i}∼N\left(0,\sigma_{0}^{2}\right)$; for Set 5, we have that $\epsilon_{i}∼t_{df=5}$, where $t_{df}$ denotes the Student’s *t* distribution based on df degrees of freedom; and for Set 6, we have that $\epsilon_{i}∼Z\cdot N\left(0,1\right)+\left(1-Z\right)\cdot N\left(0,50\right)$, where Z∼Bernoulli(π) with π=0.85.

In the setting at hand, the true data-generating model *g* has parameters $\theta ={\left(\beta_{0}^{T},\sigma_{0}^{2}\right)}^{T}$. Hurvich and Tsai [6] showed that for the family of approximating models y=Xβ+ϵ, where *X* is the design matrix and $\epsilon ∼N(0,\sigma^{2}I_{n})$, with maximum likelihood estimators given by
$$\hat{\beta }={\left(X^{T}X\right)}^{-1}X^{T}y$$
and
$${\hat{\sigma }}^{2}=\frac{{\left(y-X\hat{\beta }\right)}^{T}\left(y-X\hat{\beta }\right)}{n},$$
the KLD measure $d(g,\hat{\theta })$ is given by
(9)$$d\left(g,\hat{\theta }\right)=n\log\left(2\pi {\hat{\sigma }}^{2}\right)+\frac{n\sigma_{0}^{2}}{{\hat{\sigma }}^{2}}+\frac{{\left(X\beta_{0}-X\hat{\beta }\right)}^{T}\left(X\beta_{0}-X\hat{\beta }\right)}{{\hat{\sigma }}^{2}}.$$

The expected value of the KLD for the null and the alternative models was approximated by averaging the KLD over 5000 samples generated from *g*. These 5000 KLD values, computed using (Equation 9), approximate the joint distribution of $d(g,{\hat{\theta }}_{1})$ and $d(g,{\hat{\theta }}_{2})$; hence, the simulation-based estimator of the KLDCP is given by
(10)$$\hat{P}=\frac{1}{5000}\sum_{i=1}^{5000}I\left[d\left(g,{\hat{\theta }}_{1}\left(i\right)\right)<d\left(g,{\hat{\theta }}_{2}\left(i\right)\right)\right].$$

This KLDCP estimate is calculated 100 times in order to estimate the KLDCP distribution and its expected value.

Finally, for each of the 5000 samples, we calculate the BD and the BDb using 200 bootstrap samples. However, to attenuate the simulation variability incurred by the mixture distribution, the number of bootstrap samples in Set 6 was increased to 500. The results displayed in the tables are based on averages over the 5000 samples.

Set 1: *Null hypothesis is correctly specified, and alternative hypothesis is overspecified.*

Consider the true data-generating model given by
$$y_{i}=\beta_{0,1}+\beta_{0,2}x_{i2}+\beta_{0,3}x_{i3}+\epsilon_{i},$$
where $\epsilon_{i}∼N\left(0,50\right)$, $\beta_{0,1}=1$, $\beta_{0,2}=\beta_{0,3}=0.5$ and ${\left[\begin{array}{cc} x_{i2} & x_{i3} \end{array}\right]}^{T}$ is sampled as indicated in (Equation 8).

For the hypothesis testing setting in Set 1, the null and alternative models are defined as
$$\begin{array}{cc} \; & H_{1}:y_{i}=\beta_{1}+\beta_{2}x_{2i}+\beta_{3}x_{i3},\\ \; & H_{2}:y_{i}=\beta_{1}+\beta_{2}x_{i2}+\beta_{3}x_{i3}+\beta_{4}x_{i4}+\beta_{5}x_{i5}+\beta_{6}x_{i6}+\beta_{7}x_{i7}. \end{array}$$

Note that the null model is adequately specified, while the alternative model contains the true model plus four additional explanatory variables. These extra explanatory variables are generated from the distribution indicated in (Equation 8).

Set 2: *Null hypothesis is underspecified, and alternative hypothesis is correctly specified.*

Consider the true data-generating model given by
$$y_{i}=\beta_{0,1}+\beta_{0,2}x_{i2}+\beta_{0,3}x_{i3}+\beta_{0,4}x_{i4}+\beta_{0,5}x_{i5}+\epsilon_{i},$$
where $\epsilon_{i}∼N\left(0,45\right)$, $\beta_{0,1}=1,\beta_{0,2}=0.11,\beta_{0,3}=0.13,\beta_{0,4}=0.12,\beta_{0,5}=-0.11$, and ${\left[\begin{array}{cccc} x_{i2} & x_{i3} & \cdots & x_{i5} \end{array}\right]}^{T}$ is sampled as indicated in (Equation 8).

For the hypothesis testing setting in Set 2, the null and alternative models are
$$\begin{array}{cc} \; & H_{1}:y_{i}=\beta_{1}+\beta_{2}x_{2i}+\beta_{3}x_{i3}+\beta_{4}x_{i4},\\ \; & H_{2}:y_{i}=\beta_{1}+\beta_{2}x_{i2}+\beta_{3}x_{i3}+\beta_{4}x_{i4}+\beta_{5}x_{i5}. \end{array}$$

Here, the alternative model has the same structure as the data-generating model, but the null model is missing one of the explanatory variables in the true model, namely $x_{5}.$

Set 3: *Both null and alternative models are underspecified, but the null is closer to the data-generating model.*

Consider the true data-generating model given by
$$y_{i}=\beta_{0,1}+\beta_{0,2}x_{i2}+\beta_{0,3}x_{i3}+\beta_{0,4}x_{i4}+\beta_{0,5}x_{i5}+\beta_{0,6}x_{i6}+\epsilon_{i},$$
where $\epsilon_{i}∼N\left(0,50\right)$, $\beta_{0,1}=1,\beta_{0,2}=\beta_{0,3}=0.5,\beta_{0,4}=\beta_{0,5}=-0.5,\beta_{0.6}=0.1$, and ${\left[\begin{array}{cccc} x_{i2} & x_{i3} & \cdots & x_{i6} \end{array}\right]}^{T}$ is sampled as indicated in (Equation 8).

For the hypothesis testing setting in Set 3, the null and alternative models are
$$\begin{array}{cc} \; & H_{1}:y_{i}=\beta_{1}+\beta_{2}x_{2i}+\beta_{3}x_{i3},\\ \; & H_{2}:y_{i}=\beta_{1}+\beta_{4}x_{i4}+\beta_{6}x_{i6}. \end{array}$$

In this setting, both the null and alternative candidate models have the same number of explanatory variables, and they are both missing variable $x_{4}$. However, there is a slight difference in the effect sizes of the variables for these models. For the alternative, the effect sizes are −0.5 and 0.1 for $x_{4}$ and $x_{6}$, respectively. On the other hand, the effect size for the null model is 0.5 for both $x_{2}$ and $x_{3}$. When comparing the null and alternative models, the smaller effect size on $x_{6}$ sets the alternative further away from the true model.

Set 4: *Both null and alternative models are equally underspecified.*

Consider the true data-generating model given by
$$y_{i}=\beta_{0,1}+\beta_{0,2}x_{i2}+\beta_{0,3}x_{i3}+\beta_{0,4}x_{i4}+\beta_{0,5}x_{i5}+\beta_{0,6}x_{i6}+\beta_{0,7}x_{i7}+\epsilon_{i},$$
with $\epsilon_{i}∼N\left(0,50\right)$, $\beta_{0,1}=1,\beta_{0,2}=\beta_{0,3}=\beta_{0,6}=\beta_{0,7}=0.5,\beta_{0,4}=\beta_{0,5}=-0.5$, and ${\left[\begin{array}{cccc} x_{i1} & x_{i2} & \cdots & x_{i7} \end{array}\right]}^{T}$ is sampled as indicated in (Equation 8).

For the hypothesis testing setting in Set 4, the null and alternative models are
$$\begin{array}{cc} \; & H_{1}:y_{i}=\beta_{1}+\beta_{2}x_{2i}+\beta_{3}x_{i3},\\ \; & H_{2}:y_{i}=\beta_{1}+\beta_{4}x_{i4}+\beta_{5}x_{i5}. \end{array}$$

Here, the null and alternative candidate models are equally underspecified because they have the same number of explanatory variables with the same effect sizes, and neither model captures the true data-generating model.

Set 5: *Null model has correct mean specification and alternative model is overspecified, but both are misspecified with respect to the error distribution, which is a Student’s t distribution.*

Consider the true data generating model given by
$$y_{i}=\beta_{0,1}+\epsilon_{i},$$
with $\epsilon_{i}∼t_{df=5}$ and $\beta_{0,1}=1$. Therefore, $\sigma_{0}^{2}=\frac{5}{3}$.

For the hypothesis testing setting in Set 5, the null and alternative models are
$$\begin{array}{cc} \; & H_{1}:y_{i}=\beta_{1},\\ \; & H_{2}:y_{i}=\beta_{1}+\beta_{2}x_{i2}, \end{array}$$
where $x_{i2}∼N\left(1,100\right)$. This setting is similar to the one displayed in Set 1, where the null is properly specified while the alternative is overspecified. However, the models in the setting at hand inadequately specify the distribution of the errors.

Set 6: *Null model has correct mean specification, and the alternative model is overspecified, but both are misspecified with respect to the error distribution, which is a mixture of normals.*

Consider the true data-generating model given by
$$y_{i}=\beta_{0,1}+\epsilon_{i},$$
with $\epsilon_{i}∼Z\cdot N\left(0,1\right)+\left(1-Z\right)\cdot N\left(0,50\right)$, where Z∼Bernoulli(π) with π=0.85. Therefore,
$$\begin{array}{cc} \sigma_{0}^{2} & =0.85(1)+0.15(50)\\ \; & =8.35. \end{array}$$

For the hypothesis testing setting in Set 6, the null and alternative models are
$$\begin{array}{cc} \; & H_{1}:y_{i}=\beta_{1},\\ \; & H_{2}:y_{i}=\beta_{1}+\beta_{2}x_{i2}, \end{array}$$
where $x_{i2}∼N\left(1,100\right)$. This setting is similar to the one featured in Set 5. However, the errors in the setting at hand are generated from a mixture of normal distributions.

### 4.2. KLDCP Estimates from Simulations

For the tables showing the KLDCP simulation results, the columns are labeled as follows.
(1)**KLDCP** corresponds to results based on the distribution of 100 replicates of KLDCP, where each KLDCP is calculated using (Equation 10). Note that the null and alternative KLD joint distribution is characterized based on discrepancy replicates obtained through (Equation 9).(2)**BDCPb** corresponds to results based on the distribution of 5000 replicates of BDCPb. Each BDCPb is computed using (Equation 6) with 200 bootstrap samples for Sets 1–5 and 500 bootstrap samples for Set 6.(3)**BDCPk** corresponds to results based on the distribution of 5000 replicates of BDCPk. Each BDCPk is computed using (Equation 7) with 200 bootstrap samples for Sets 1–5 and 500 bootstrap samples for Set 6.(4)**BDCP** corresponds to results based on the distribution of 5000 replicates of the uncorrected BDCP. Each BDCP is computed using (Equation 1) with 200 bootstrap samples for Sets 1–5 and 500 bootstrap samples for Set 6.

### 4.3. Estimates of the Expected KLD from Simulations

For the tables showing the KLD results, the columns are labeled as follows.
(1)**E(KLD)** corresponds to the average of 5000 discrepancies calculated using (Equation 9).(2)**E(BD)** corresponds to the average of 5000 replicates of BD, where each BD is calculated by
$$\frac{1}{M}\sum_{m=1}^{M}-2\ell \left({\hat{\theta }}^{*}\left(m\right)|y\right).$$We have that M=200 for Sets 1–5 and M=500 for Set 6.(3)Δ**BDb** corresponds to the difference between the estimate of *E*(BD), with each BD corrected by $k_{b}$ and the estimate of *E*(KLD) described in (1). In other words, if we let j∈{1,2…,5000} be the number of simulated data sets, ${\tilde{BD}}_{j}$ be the BD estimate for each data set *j*, and $k_{jb}$ be the $k_{b}$ correction for data set *j*, then
$$\Delta \mathrm{BDb}=\frac{1}{5000}\sum_{j=1}^{5000}\left[{\tilde{BD}}_{j}+k_{jb}\right]-E\left(KLD\right).$$(4)Δ**BDk** shows the same difference described in (3), but using *k* instead of $k_{b}$, which results in
$$\Delta \mathrm{BDk}=\frac{1}{5000}\sum_{j=1}^{5000}\left[{\tilde{BD}}_{j}+k\right]-E\left(KLD\right).$$

### 4.4. Discussion of Simulation Results

As mentioned previously, in the conventional hypothesis testing scenario for comparing nested models, Riedle, Neath and Cavanaugh [1] established that the uncorrected BDCP approximates the *p*-value derived from the likelihood ratio test. Therefore, in the case where the null candidate model is correctly specified, both the uncorrected BDCP and the *p*-value have a Uniform(0,1) distribution. This behavior is displayed in Table 1, where for large sample sizes, the mean and median of the BDCP distribution are around 0.5. This is a problematic feature of the uncorrected BDCP and *p*-values because the measure does not reliably favor the null model in those settings where the null is true. However, we see that for large sample sizes, both the BDCPk and the BDCPb values are close to 1, which clearly favors the null model.

Table 2 shows the results from the setting where the alternative hypothesis is correctly specified, while the null is underspecified. Here, we would expect all the discrepancy probabilities to be close to 0, as seen in the case where the sample size is N=500. However, for smaller sample sizes, i.e., N=25 and N=50, we observe larger values for the discrepancy probabilities. In fact, for N=25, the BDCPb is 0.89 and, with a mean and median close to 0.5, the uncorrected BDCP exhibits similar behavior to the case where the null is true. This phenomenon is expected within the framework of model selection, where additional explanatory variables are favorable if there is a sufficient sample size to adequately estimate their effects. If the sample size is too small to construct reliable estimates, then it is best to choose smaller models, even at the expense of model misspecification.

The results from Table 1, Table 3, Table 4, Table 5 and Table 6 show that when estimating the KLDCP with a small sample size (N=25 to N=100), the BDb performs either better than or as well as the BDk. For large sample sizes, all simulation sets exhibit a similar performance for both corrections.

For discrepancy estimation, Table 7, Table 8, Table 9 and Table 10 show that across all sample sizes, $k_{b}$ over-corrects for the bias of the discrepancy approximation, and the over correction is more prominent for small sample sizes. It is worth noting that this evident over-estimation from the BDb is accompanied by a superior bias reduction of the corresponding KLDCP estimator. For instance, Table 7 shows a significant over-estimation by BDb compared to BDk, especially in the small sample settings. However, the corresponding estimator of the KLDCP, displayed in Table 1, exhibits less bias for BDCPb than for BDCPk.

Finally, Table 11 and Table 12 show that, across all sample sizes, the correction by $k_{b}$ markedly reduces the bias compared to the correction by *k*. This means that in the setting where the mean structure is correctly specified for the null and overspecified for the alternative, but both models are incorrectly specified with respect to the error distribution, the bootstrap-based correction evidently outperforms the simple correction of *k*.

In most cases, however, the bias reductions resulting from the $k_{b}$ and the *k* corrections are comparable. Therefore, our simulation studies suggest that if the null and/or the alternative models are misspecified, then correcting by either $k_{b}$ or *k* will generally yield comparable estimators of the expected KLDCP.

## 5. Application: Creatine Kinase Levels during Football Preseason

In this section, we apply the BDCP to a data set from a biomedical setting. The goal of this application is to understand the changes in creatine kinase (CK) levels observed on the blood samples of college football players during preseason training. In order to properly explain the variation of CK, we must select between competing models that use different demographic and clinical variables. We will analyze the models selected by the $k_{b}$ corrected, the *k* corrected and the uncorrected BDCP, and we will compare the results to the selection of models via the more conventional *p*-value approach.

### 5.1. Overview of Application

During strenuous exercise, skeletal muscle cells break down and release a variety of intracellular contents. When in excess, a condition known as exertional rhabdomyolysis (ER) can occur, which may result in life-threatening complications such as renal failure, cardiac arrhythmia and compartment syndrome. Creatine kinase (CK) is one of the proteins released during muscle breakdown, and measuring its levels is the most sensitive test for assessing muscular damage that could lead to ER [7].

During the off-season workouts in January 2011, a group of 13 University of Iowa football players developed ER. This event led to a prospective study where 30 University of Iowa football athletes were followed during a 34-day preseason workout camp. Variables such as body mass index (BMI) and CK levels were obtained from blood samples that were drawn at the first, third, and seventh day of the camp. Other demographic and clinical variables such as age, number of semesters in the program and history of rhabdomyolysis were also collected.

The initial results of the study, published by Smoot et al. [8], show that the CK levels at later time points were significantly different than the levels at earlier times. However, most of the clinical and demographic variables were not significant in explaining the levels of CK. One of the underlying issues with this type of modeling analysis is that the significance of each variable can only be assessed by hypothesis tests with nested models. For example, suppose that we wish to determine the significance of BMI in the presence of semesters in the program. To obtain a *p*-value for BMI, we need to formulate a hypothesis test where the null model only contains semesters in the program, while the alternative model contains both BMI and semesters in the program.

Although this setting may be useful in some scenarios, it is too limiting. For instance, suppose that we wish to choose between two non-nested models where one contains BMI and the other contains semesters in the program. Although a conventional test based on linear regression models would not be able to answer this question, the BDCP approach could indeed determine the propriety of either model in this type of non-nested setting.

In the analysis of this data set, we let CK3 be the log of CK levels measured at the seventh day of the camp, CK1 be the log of CK levels measured at the first day of the camp, and Semesters be the number of semesters at the program. Of note, the log transformation is routinely applied in studies involving CK levels in order to justify approximate normality, as the raw levels tend to have heavily right-skewed distributions.

Now, consider the following hypothesis testing settings.

Setting 1: *Testing the propriety of the model containing CK1.*
$$\begin{array}{cc} \; & H_{1}:CK3=\beta_{1},\\ \; & H_{2}:CK3=\beta_{1}+\beta_{2}\;CK1. \end{array}$$Setting 2: *Testing the propriety of the model containing CK1 and Semesters over the model containing only CK1.*
$$\begin{array}{cc} \; & H_{1}:CK3=\beta_{1}+\beta_{2}\;CK1,\\ \; & H_{2}:CK3=\beta_{1}+\beta_{2}\;CK1+\beta_{3}\;Semesters. \end{array}$$Setting 3: *Head-to-head comparison of non-nested models.*
$$\begin{array}{cc} \; & H_{1}:CK3=\beta_{1}+\beta_{2}\;CK1+\beta_{3}\;BMI,\\ \; & H_{2}:CK3=\beta_{1}+\beta_{2}\;CK1+\beta_{3}\;Semesters. \end{array}$$

### 5.2. Results of Application

The results for the application are summarized in Table 13. Settings 1 and 2 illustrate the congruence between BDCP and *p*-values in the case of hypothesis testing based on nested models. Setting 1 assesses the propriety of a model that includes only the intercept against a model that includes both the intercept and the levels of CK1. The *p*-value for CK1 in this setting is 0.001, which means that, using a level α of 0.05, CK1 is significant in explaining the variation in CK3 levels. Both the BDCPk and BDCPb are 0.075, which means that there is a 7.5% chance that the null model is preferred over multiple bootstrap samples, indicating that the model containing CK1 is superior.

Once we establish that CK1 is an important variable to include in our model, the next step is to determine if additional variables can improve our model fit. Setting 2 displays a hypothesis test where the null model only contains CK1, while the alternative contains both CK1 and Semesters. The *p*-value for Semesters is 0.734, which means that Semesters is not statistically significant, and a reasonable investigator would choose to exclude Semesters from the final model. The corrected BDCP values arrive at the same conclusion. For instance, the BDCPb is 0.995, which indicates that the across multiple bootstrap samples, the null model is chosen 99.5% of the time; therefore, the BDCP encourages us to choose the model that excludes Semesters.

The rationale for testing Semesters is based on the idea that more senior athletes tend to rigorously maintain their workout habits during the off season, mostly because of experience and maturity. Therefore, Semesters is a variable that may confound the effects of CK1 on the variation of CK3. Additionally, medical literature has shown that BMI highly correlates with CK levels and the development of ER [9], which means that one should also test for the propriety of models that include BMI. Thus, one could ask if a model featuring BMI would be better than a model featuring Semesters. This results in a hypothesis testing scenario where the null and alternative models are non-nested, as exhibited in Setting 3.

First, note that the *p*-values displayed in the table for Setting 3 do not answer the question at hand. These *p*-values are obtained from partial tests applied to the full model containing both variables. On the other hand, the BDCP gives us meaningful information about the performance of adding BMI versus adding Semesters. The BDCPb tells us that there is a 78% probability that the model containing BMI is a better fit than the model containing Semesters. If we use the BDCPk instead, the probability increases to 81.5%. In both cases, if we are debating weather to include BMI or Semesters as an adjusting variable, the BDCP clearly favors the inclusion of BMI.

## 6. Conclusions

When deciding between two competing models, practitioners of statistics normally utilize traditional hypothesis testing methods that rely on the assumption that one of the candidate models is properly specified. This approach is problematic because it is unreasonable to assume that one of the proposed models is precisely true. In addition, these methods are only applicable for nested models. To avoid any underlying assumptions and model structure limitations, Riedle, Neath and Cavanaugh [1] propose the use of the bootstrap discrepancy probability (BDCP) to assess the propriety of the fit of two candidate models. However, the bootstrap discrepancy (BD) utilized in this work provides a biased estimator of the Kullback–Leibler discrepancy (KLD).

When hypothesis testing assumptions are met, the BDCP asymptotically approximates the likelihood ratio test *p*-value. Therefore, similarly to *p*-values, the distribution of the BDCP is uniform if the null hypothesis is true. Hence, in settings when the null is true, the BDCP would be of limited value in choosing the appropriate model.

In this paper, we proposed utilizing the $k_{b}$ or the *k* corrected BDCP, namely BDCPb and BDCPk, respectively. The BDCPb employs the BDb, a bootstrap corrected estimator of the KLD, while the BDCPk uses the BDk, a BD corrected by adding the number of functionally independent parameters in the candidate model. We showed that for most settings, the BDb serves as an over-corrected estimator of the KLD, but the corresponding BDCPb is less biased than the BDCPk for the estimation of the KLDCP. However, in the case when there is distributional misspecification, we showed that the BDb has negligible bias for the estimation of expected value of the KLD.

Moreover, the estimation of the bootstrap correction $k_{b}$ utilizes the same bootstrap samples that were used to calculate the BD; therefore, we argue that the computational requirements of estimating $k_{b}$ are not too burdensome. However, if the sample size is moderately large compared to the number of parameters in the model, then we showed that using *k* to correct the bias generally results in comparable values of the KLDCP estimates.

## Figures and Tables

**Table 1 entropy-24-01483-t001:** Distribution approximations for Set 1, where the null model is correctly specified, while the alternative model is overspecified.

Statistic	KLDCP	BDCPb	BDCPk	BDCP
**N = 500**				
Mean	1.000	0.878	0.868	0.515
Median	1.000	1.000	1.000	0.515
SD	0.000	0.233	0.241	0.282
**N = 100**				
Mean	1.000	0.918	0.864	0.564
Median	1.000	1.000	0.995	0.580
SD	0.000	0.186	0.225	0.256
**N = 50**				
Mean	1.000	0.966	0.875	0.631
Median	1.000	1.000	0.980	0.650
SD	0.000	0.111	0.193	0.220
**N = 25**				
Mean	1.000	0.999	0.886	0.739
Median	1.000	1.000	0.955	0.755
SD	0.000	0.012	0.144	0.156

**Table 2 entropy-24-01483-t002:** Distribution approximations for Set 2, where the null model is underspecified, while the alternative model is correctly specified.

Statistic	KLDCP	BDCPb	BDCPk	BDCP
**N = 500**				
Mean	0.001	0.022	0.021	0.011
Median	0.001	0.000	0.000	0.000
SD	0.000	0.088	0.085	0.043
**N = 100**				
Mean	0.156	0.470	0.428	0.264
Median	0.156	0.340	0.280	0.170
SD	0.005	0.390	0.378	0.257
**N = 50**				
Mean	0.372	0.691	0.597	0.409
Median	0.372	0.905	0.630	0.360
SD	0.007	0.350	0.354	0.266
**N = 25**				
Mean	0.617	0.890	0.698	0.536
Median	0.617	0.990	0.785	0.535
SD	0.006	0.213	0.280	0.222

**Table 3 entropy-24-01483-t003:** Distribution approximations for Set 3, where the null and alternative models are underspecified, but the null model is closer to the true data-generating model.

Statistic	KLDCP	BDCPb	BDCPk	BDCP
**N = 500**				
Mean	1.000	1.000	1.000	1.000
Median	1.000	1.000	1.000	1.000
SD	0.000	0.013	0.013	0.013
**N = 100**				
Mean	0.979	0.910	0.910	0.910
Median	0.979	1.000	1.000	1.000
SD	0.002	0.244	0.244	0.244
**N = 50**				
Mean	0.916	0.807	0.808	0.808
Median	0.916	0.970	0.970	0.970
SD	0.004	0.311	0.309	0.309
**N = 25**				
Mean	0.804	0.692	0.699	0.699
Median	0.805	0.845	0.840	0.840
SD	0.005	0.314	0.303	0.303

**Table 4 entropy-24-01483-t004:** Distribution approximations for Set 4, where the null and alternative models are equally underspecified.

Statistic	KLDCP	BDCPb	BDCPk	BDCP
**N = 500**				
Mean	0.498	0.507	0.507	0.507
Median	0.498	0.570	0.580	0.580
SD	0.007	0.478	0.478	0.478
**N = 100**				
Mean	0.500	0.510	0.509	0.509
Median	0.500	0.562	0.567	0.567
SD	0.007	0.442	0.442	0.442
**N = 50**				
Mean	0.500	0.502	0.502	0.502
Median	0.500	0.505	0.515	0.515
SD	0.007	0.407	0.406	0.406
**N = 25**				
Mean	0.501	0.501	0.501	0.501
Median	0.501	0.490	0.495	0.495
SD	0.007	0.353	0.345	0.345

**Table 5 entropy-24-01483-t005:** Distribution approximations for Set 5, where the null and alternative models are misspecified with respect to the error distribution. Here, the errors are generated from a Student’s t distribution.

Statistic	KLDCP	BDCPb	BDCPk	BDCP
**N = 500**				
Mean	1.000	0.794	0.794	0.499
Median	1.000	1.000	1.000	0.500
SD	0.000	0.329	0.328	0.289
**N = 100**				
Mean	1.000	0.807	0.794	0.507
Median	1.000	1.000	1.000	0.515
SD	0.000	0.318	0.323	0.284
**N = 50**				
Mean	1.000	0.825	0.790	0.508
Median	1.000	1.000	0.995	0.505
SD	0.000	0.301	0.315	0.273
**N = 25**				
Mean	1.000	0.862	0.790	0.525
Median	1.000	1.000	0.985	0.530
SD	0.000	0.270	0.306	0.261

**Table 6 entropy-24-01483-t006:** Distribution approximations for Set 6, where the null and alternative models are misspecified with respect to the error distribution. Here, the errors are generated from a mixture of normal distributions.

Statistic	KLDCP	BDCPb	BDCPk	BDCP
**N = 500**				
Mean	1.000	0.783	0.786	0.487
Median	1.000	1.000	1.000	0.484
SD	0.000	0.338	0.335	0.289
**N = 100**				
Mean	1.000	0.808	0.793	0.495
Median	1.000	1.000	0.998	0.496
SD	0.000	0.322	0.325	0.283
**N = 50**				
Mean	1.000	0.851	0.793	0.502
Median	1.000	1.000	0.994	0.494
SD	0.000	0.286	0.311	0.269
**N = 25**				
Mean	1.000	0.906	0.787	0.509
Median	1.000	1.000	0.986	0.490
SD	0.000	0.229	0.300	0.246

**Table 7 entropy-24-01483-t007:** Expected value of the KLD, its bootstrap estimate, and the bias of the corrected bootstrap estimates for the null and alternative models in Set 1. Here, the null model is correctly specified, while the alternative model is overspecified.

Hypothesis	*E*(KLD)	*E*(BD)	ΔBDb	ΔBDk
**N = 500**				
Null	3378.949	3375.407	0.488	0.411
Alternative	3383.138	3375.578	0.686	0.362
**N = 100**				
Null	679.282	675.291	0.385	−0.030
Alternative	684.115	676.667	2.518	0.521
**N = 50**				
Null	342.167	338.498	1.267	0.268
Alternative	348.245	342.348	7.476	2.065
**N = 25**				
Null	174.334	171.169	3.657	0.910
Alternative	183.828	193.249	43.328	17.290

**Table 8 entropy-24-01483-t008:** Expected value of the KLD, its bootstrap estimate, and the bias of the corrected bootstrap estimates for the null and alternative models in Set 2. Here, the null model is underspecified, while the alternative model is correctly specified.

Hypothesis	*E*(KLD)	*E*(BD)	ΔBDb	ΔBDk
**N = 500**				
Null	3340.491	3335.733	0.410	0.290
Alternative	3328.467	3322.581	0.319	0.143
**N = 100**				
Null	672.373	667.928	1.210	0.520
Alternative	671.137	665.628	1.493	0.454
**N = 50**				
Null	339.515	334.726	1.891	0.226
Alternative	339.923	334.181	2.888	0.305
**N = 25**				
Null	174.136	171.376	7.446	2.223
Alternative	176.073	174.320	13.270	4.106

**Table 9 entropy-24-01483-t009:** Expected value of the KLD, its bootstrap estimate, and the bias of the corrected bootstrap estimates for the null and alternative models in Set 3. Here, the null and alternative models are underspecified, but the null model is closer to the true data-generating model.

Hypothesis	*E*(KLD)	*E*(BD)	ΔBDb	ΔBDk
**N = 500**				
Null	3726.902	3726.159	3.401	3.332
Alternative	3832.770	3832.395	3.704	3.626
**N = 100**				
Null	745.967	745.809	4.358	3.943
Alternative	766.212	766.813	4.947	4.528
**N = 50**				
Null	373.419	373.704	5.309	4.325
Alternative	383.156	384.020	5.843	4.858
**N = 25**				
Null	187.563	188.745	8.082	5.245
Alternative	191.924	194.082	8.878	6.088

**Table 10 entropy-24-01483-t010:** Expected value of the KLD, its bootstrap estimate, and the bias of the corrected bootstrap estimates for the null and alternative models in Set 4. Here, the null and alternative models are equally underspecified.

Hypothesis	*E*(KLD)	*E*(BD)	ΔBDb	ΔBDk
**N = 500**				
Null	3923.423	3923.908	5.022	4.948
Alternative	3923.580	3924.705	5.475	5.399
**N = 100**				
Null	784.021	784.917	5.080	4.670
Alternative	784.042	785.026	5.241	4.823
**N = 50**				
Null	391.751	393.155	6.335	5.343
Alternative	391.753	393.131	6.222	5.239
**N = 25**				
Null	195.732	198.616	9.602	6.821
Alternative	195.862	198.690	9.598	6.804

**Table 11 entropy-24-01483-t011:** Expected value of the KLD, its bootstrap estimate, and the bias of the corrected bootstrap estimates for the null and alternative models in Set 5. Here, the null and alternative models are misspecified with respect to the error distribution, and the errors are generated from a Student’s t distribution.

Hypothesis	*E*(KLD)	*E*(BD)	ΔBDb	ΔBDk
**N = 500**				
Null	1678.652	1672.369	−2.224	−4.178
Alternative	1679.695	1672.387	−2.248	−4.231
**N = 100**				
Null	338.728	334.154	−0.920	−2.471
Alternative	339.866	334.300	−0.728	−2.438
**N = 50**				
Null	171.377	167.500	−0.231	−1.839
Alternative	172.640	167.847	0.283	−1.714
**N = 25**				
Null	87.689	83.577	−0.434	−2.077
Alternative	89.311	84.495	0.869	−1.785

**Table 12 entropy-24-01483-t012:** Expected value of the KLD, its bootstrap estimate, and the bias of the corrected bootstrap estimates for the null and alternative models in Set 6. Here, the null and alternative models are misspecified with respect to the error distribution, and the errors are generated from a mixture of normal distributions.

Hypothesis	*E*(KLD)	*E*(BD)	ΔBDb	ΔBDk
**N = 500**				
Null	2488.932	2480.154	−0.389	6.554
Alternative	2490.012	2480.141	−0.310	6.659
**N = 100**				
Null	508.122	497.000	−0.383	8.404
Alternative	509.426	497.237	−0.597	8.459
**N = 50**				
Null	263.382	252.424	−2.852	8.590
Alternative	264.974	253.245	−3.930	8.361
**N = 25**				
Null	144.895	131.870	−4.361	10.842
Alternative	147.551	134.298	−7.782	9.956

**Table 13 entropy-24-01483-t013:** From left to right: results for Setting 1, Setting 2, and Setting 3. BDCPk is the BDCP corrected by *k*, BDCPb is the BDCP corrected by $k_{b}$, and BDCP is the uncorrected BDCP. Results are based on 200 bootstraps samples.

BDCP					
BDCPk	0.075	BDCPk	0.990	BDCPk	0.815
BDCPb	0.075	BDCPb	0.995	BDCPb	0.780
BDCP	0.055	BDCP	0.495	BDCP	0.815
* **p** * **-Value**					
CK1	0.001	CK1	0.001	CK1	0.001
		Semesters	0.734	BMI	0.176
				Semesters	0.936

## Data Availability

The R code used in generating the data for the simulation study is available on request from the corresponding author. The data for the application are not publicly available since the dataset is confidential.

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
