# Peer review of "Probabilistic Pairwise Model Comparisons Based on Bootstrap Estimators of the Kullback–Leibler Discrepancy"

_entropy, 2022, doi:10.3390/e24101483_

Round 1
Reviewer 1 Report
Authors in this paper use a bootstrap approximation of the Kullback-Leibler discrepancy to estimate the probability of fitting the model.They propose correction of the bias.
In the simulation studies they used a maximum likelihood estimators and they conduct testing.
Next for n = 25, 50, 100 and 500 they give a mean and median for all used methods.
Finaly they give results of expected value for bootstrap estimate and bias of the corrected bootstrap.
Then they discuss the simulation and give applications.
The obtained results show that the authors managed to introduce a model that functions better than the previous ones.
Therefore I recommend the paper by Dajles A. and Cavanaugh J. entitled “Probabilistic Pairwise Model Comparisons Based on Bootstrap Estimators of the Kullback-Leibler Discrepancy” for publication in “Entropy” journal.
Reviewer 2 Report
See the attached report.
